# Emergent Communication in Interactive Sketch Question Answering

**Zixing Lei**[1], **Yiming Zhang**[2], **Yuxin Xiong**[1], **Siheng Chen**[1,3]*
chezacarss@sjtu.edu.cn; yz2926@cornell.edu;
xyx1323@sjtu.edu.cn; sihengc@sjtu.edu.cn
[1] Cooperative Medianet Innovation Center, Shanghai Jiao Tong University,
[2] Cornell University, [3] Shanghai AI Laboratory

Figure 1: In the proposed interactive sketch question answering (ISQA) task, two collaborative players are interacting to answer a question about an image. This task emphasizes multi-round interaction, which is essential in daily human communication.

## Abstract

Vision-based emergent communication (EC) aims to learn to communicate through sketches and demystify the evolution of human communication. Ironically, previous works neglect multi-round interaction, which is indispensable in human communication. To fill this gap, we first introduce a novel Interactive Sketch Question Answering (ISQA) task, where two collaborative players are interacting through sketches to answer a question about an image in a multi-round manner. To accomplish this task, we design a new and efficient interactive EC system, which can achieve an effective balance among three evaluation factors, including the question answering accuracy, drawing complexity and human interpretability. Our experimental results including human evaluation demonstrate that multi-round interactive mechanism facilitates targeted and efficient communication between intelligent agents with decent human interpretability. The code is available at here.

## 1 Introduction

Emergent communication aims for a better understanding of human language evolution and is a promising direction for achieving human-like communication between intelligent agents [1, 2, 3, 4]. An ultimate EC system allows intelligent agents to interact and exchange information with one another, facilitating collaboration in the completion of downstream tasks. EC is expected to play an important role in a wide range of applications, including multi-robot navigation [5], collaborative perception [6] and human-centric AI applications [7].

Previous methods on EC can be divided into language-based EC and vision-based EC. Language-based EC aims to encode information into message vectors or nature language for transmission between intelligent agents [1, 2, 3, 8, 9, 10, 11, 12]. Vision-based EC systems have been developed with the aim of imitating the pictographic systems that were used in early human communication, and have shown promising results in achieving human interpretability [13, 14]. A representative vision-based EC approach [13] is inspired by the popular game "Draw & Guess," where the sender acts as a sketcher while the receiver is responsible for making guesses. Sketch is a visible and

---
*Corresponding author

37th Conference on Neural Information Processing Systems (NeurIPS 2023).

universal form to transmit information, which is suitable for building human-interpretable systems. Following this spirit, this work specifically focuses on sketch-based EC.

In this emerging direction, current works on sketch-based EC show two critical issues. First, from the aspect of task setting, multi-round interaction is underexplored. As a fundamental aspect of human communication, interaction involves ongoing feedback and adjustment, facilitating bidirectional communication between intelligent agents. However, previous works on EC largely focused on single-round communication for the tasks of image classification or Lewis's game [15]. A recent work [14] proposes a multi-round mechanism, but its receiver is still limited to producing binary flags, indicating whether or not it is confident of its decision. This plain feedback cannot provide sufficient feedback for the sender to promote meaningful interaction. Second, in terms of evaluation, previous works on sketch-based EC primarily focus on optimizing for the performances of downstream tasks without sufficient consideration of communication efficiency and quantitive interpretability. This would result in complex and non-interpretable information interchange, which is not suitable for human-centric AI systems.

To address the task setting issue, we propose a new multi-round interactive task, named interactive sketch question answering (ISQA); see an illustration in Fig. 1. In this task, there are two players: a sender and a receiver. An RBG image is provided to the sender and the question related to this image is given to the receiver. Based on the image, the sender can generate a sketch and transmit it to the receiver with certain drawing complexity constraints. According to the sketch, the receiver generates an answer to the question. If the answer is wrong, the receiver can send spatial bounding boxes to the sender as feedback, which indicate which parts of the sketch require more detailed drawing in subsequent rounds. Through multiple rounds of interaction, the receiver gains more information about the image and achieves better question answering performance. Compared to previous tasks, the new task of ISQA calls more necessity of interaction because it creates a game of incomplete information for both paricipants, where the sender does not have access to the language question and the receiver does not know the original image. This setting prompts them to engage in bidirectional information exchange through a multi-round interaction.

To address the evaluation issue, we develop a comprehensive set of metrics, covering the task performance, drawing complexity and human interpretability, to evaluate the performance in the ISQA tasks. The system's task performance is assessed by measuring its accuracy in performing ISQA tasks. The system's drawing complexity is determined by analyzing the amount of pixels the sender transmits and digits used in the interactive feedback. The system's human interpretability is determined by the CLIP distance metric between sketch and input image. By integrating these three metrics, a comprehensive evaluation of the EC system's effectiveness can be established.

Based on the ISQA task and the corresponding metrics, we propose and evaluate a novel EC system, which consists of a sender module and a receiver module. The sender module is a variable drawing complexity sketcher, and the receiver includes an SQA module and a feedback module which selects region of high relevance about the question in the sketch. The SQA module is specifically trained to adopt sketches and the feedback module provides a gradient analysis based on the answer distribution calculated by the SQA module, enabling it to select high-relevance regions in the sketch and feedback to the sender. Our experiments including human evaluation show that our EC system provides distinct performance enhancements when compared to non-interaction sketch-based EC systems, and also maintains adequate human interpretability across various complexity constraints. This validates that based on sketches, intelligent agents can emerge interactive collaboration within constrained complexity. The proposed method may pave a path to build interactive agents that can work with both human and other agents by considering others' priorities and exchanging crucial information to promote effective collaboration. Our contribution can be summarized as four aspects:
• We propose a novel interactive sketch question answering (ISQA) task for EC.
• We design an efficient and interactive EC system to improve communication quality.
• We propose a novel three-factor evaluation metric, covering question answering performance, drawing complexity and human-interpretability.
• Our experiments show that interactive EC system leverages a targeted and efficient communication.

## 2 Related work

### 2.1 Emergent communication

Emergent communication refers to the phenomenon where communication emerges among intelligent agents that share a task or incentive which can be better achieved when intelligent agents cooperate

with each other. One view of EC[16, 17] focuses on utility by framing messages to deliver high performance in downstream task. The other view inspired by a cognitive science and information theory perspective [18, 19, 20] focuses on task-agnostic communicative objectives as major forces shaping human languages [21, 22]. [23] integrates these two views of communication by optimizing a tradeoff among utility, informativeness and complexity.

Sketching is an efficient and direct way for communication. [13] designs a sender and receiver structure which allows two agents to communicate by playing "Draw & Guess" game, which requires the receiver to learn how to associate the sketch drawn by the sender with a corresponding image. However, the game is played in only one round and the receiver does not provide any information to the sender. [14] designs a multi-round communication game which allows the sender and receiver to communicate in multiple rounds to explores the change of iconicity, symbolicity, and semanticity during the iterations of training procedure. During the game, the sketch will be updated by the sender which is built upon a reinforcement learning model [24] each round until the receiver is confident to make a decision. Despite the multi-round mechanism being in place, the receiver is still constrained to output a binary flag that simply indicates whether to continue drawing or not. This cannot provide information to guide the generation of a better sketch in the subsequent round. To tackle this issue, our work focuses on multi-round interaction and the receiver can provide informative feedback.

## 2.2 Visual question answering

Visual Question Answering [25] task enables the interaction between texts and images, which has been a topic of concern in recent years. Early VQA models [26, 27, 28] often use CNN to extract image feature and RNN to embed the question. Then attention-based models [29, 30, 31] are introduced to focus on specific regions of images and questions. After transformer [32] is proposed, transformer-based models [33, 34, 35] further boost the performance of VQA models. The deep Modular Co-Attention Network (MCAN) [35] is one of the transformer-based models with high performance. It was evaluated on the benchmark VQA-v2 dataset [36] and won the VQA Challenge 2019. Our SQA model uses the structure of MCAN to predict the answer from sketches and questions.

## 2.3 Sketch generation

Drawing is a direct and efficient way for communication. The endeavor of teaching neural models to generate sketch is started by Sketch-RNN [37]. To extract more features and make the sketch more explainable, the encoder-decoder structure is used in many sketching models [38, 39]. After CLIP [40] is proposed, CLIP-guidance sketching methods [41, 42] are able to generate high-quality sketches, but the efficiency is low due to the iterative methods. [43] uses depth information, CLIP features, and the LSGAN setup [44] to train a sketch model, which achieves a good compromise between sketch quality and efficiency. Our sketch generator in the SQA game is based on [43] model.

# 3 Interactive Sketch Question Answering

## 3.1 Task formulation

In the task of interactive sketch question answering (ISQA), two collaborative players, a sender and a receiver, are interacting through sketches to answer a question about an image. The sender does not know the question and the receiver cannot see the image. The sender takes an image as input and generates a sketch with specific drawing complexity. This sketch is then transmitted to the receiver. The receiver's job is to answer a given question about the original image through the sketch from the sender. If the receiver is uncertain about the answer, it will feedback a drawing with several bounding boxes, indicating uncertain areas and requesting more detailed information about the original image from the sender. The sender, in turn, generates a new sketch according on the receiver's feedback to assist the receiver to answer the question.

Mathematically, in the $i$th interaction round, let $\mathbf{X} \in \mathcal{R}^{H \times W \times 3}$ be an RGB image, $a$ be the scalar value to indicate the human-interpretability level, $b_i$ be the scalar indicator of the drawing complexity level in the $i$th interaction round, $\mathbf{H}_i$ be the feedback provided by receiver in the $i$th round, which is initialized by zeros. Then, the sender obtains the sketch as follows,

$$\mathbf{S}_i \ = \ f_{\theta_s}\left(\mathbf{X}, \mathbf{H}_{i-1}, b_i, a\right) \in [0, 1]^{H \times W}, \tag{1}$$

where $f_{\theta_s}(\cdot)$ is the sketch generation network and $\mathbf{S}_i$ is the sketch generated in $i$th interaction round. Note that the value of each pixel in $\mathbf{S}_i$ is ranging from $0$, indicating a black and activated pixel to $1$, reflecting a white and unactivated pixel.

Correspondingly, let $\mathbf{Q}$ be a language-based question. In the $i$th interaction round, the receiver works as follow,

$$\widehat{\mathbf{A}}_i, \mathbf{H}_i \;=\; f_{\theta_r}\left(\mathbf{Q}, \{\mathbf{S}_\tau\}_{\tau=1,2,\cdots,i}\right), \tag{2}$$

where $f_{\theta_r}(\cdot)$ is the sketch question answering network, $\widehat{\mathbf{A}}_i$ is the predicted answer distribution that reflects the probability of each alternative answer, and $\mathbf{H}_i$ is a feedback sketch transmitted to sender, which consists of a set of bounding boxes indicating the spatial areas that have a high correlation with the problem.

### 3.2 Triangular evaluation

To quantify the overall performance of two collaborative players in both training and inference phases, we consider three factors: question answering ability, drawing complexity, and human interpretability.

**Question answering ability.** To reflect the performance of SQA in the training phase, we consider the binary cross-entropy (BCE) loss between the ground-truth answer $\mathbf{A}$ and the predicted answer $\widehat{\mathbf{A}}$; that is, $L_1 = \text{BCE}(\mathbf{A}, \widehat{\mathbf{A}})$. Note that the binary cross-entropy(BCE) loss is commonly used in the VQA task. In inference, we consider the asnwering accuracy following the common VQA methods.

**Drawing complexity.** To reflect the drawing complexity in both training and inference phases, we count the number of activated pixels in a sketch. When provided with a complexity constraint $b_i$, and $N$ denotes the number of pixels in original RGB image, only $b_i N$ pixels are permitted to be activated. In multi-round setting, the sender may not take full usage of $b_i$ in every round and we let $p_i$ to be the actual pixels transmitted in $i$th round and $p_i \leq b_i N$. Besides, the feedback complexity is also constrained. The complexity of feedback in $i$th round is $5h_i$, where $h_i$ denotes the numbers of bounding boxes transmitted in the $i$th round since two points and a weight can represent a weighted bounding boxes. Therefore, the total drawing complexity is $B = \sum_i (p_i + 5h_i)$.

**Human interpretability.** In the training phase, to maximize the interpretability in human vision, we can minimize the following loss,

$$L_2 = \sum_\ell \|\text{CLIP}_\ell\left(f_s\left(\mathbf{X}\right)\right) - \text{CLIP}_\ell\left(\mathbf{S}\right)\|_2^2 - \text{cos-sim}\left(\text{CLIP}\left(f_s\left(\mathbf{X}\right)\right), \text{CLIP}\left(\mathbf{S}\right)\right), \tag{3}$$

where $\text{CLIP}_\ell(\cdot)$[40] is the CLIP visual encoder activation at layer $\ell$, $\text{CLIP}(\cdot)$ is the CLIP visual encoder that outputs a feature vector, $\text{cos-sim}(\cdot)$ is the cosine similarity function and $f_s(\cdot)$ is a pre-trained sketch generation network [43]. The first term of the loss function emphasizes local geometric guidance, while the second term focuses on a semantic perspective to globally guide the sketch generation process.

In the inference phase, we also utilize $L_2$ in (3) as the evaluation criterion for human interpretability, where a lower $L_2$ loss indicates higher interpretability of the generated sketches for human perception. By integrating distance of multilevel geometric features and high-level semantic features, this metric leverages the CLIP model which is trained in 400 million (image, text) pairs and shows great zero-shot performance in vision-related task to imitate human vision.

Overall, we aim to minimize $L = L_1 + f_{\text{balance}}(a) \cdot L_2$, under various levels of complexity constraints. The hyper-parameter $a$ controls the relative weight of the $L_1$ and $L_2$ losses, determining whether the model prioritizes human interpretability or SQA accuracy. To ensure a balanced trade-off between the two loss terms, the function $f_{\text{balance}}(a)$ is introduced to adjust the weight of $L_1$ with respect to $L_2$. A default choice for $f_{\text{balance}}(a)$ is to set it to $f_{\text{balance}}(a) = 10a$.

The proposed metric enables quantitative evaluation of both accuracy and human interpretability across various drawing complexity levels. Utilizing a multi-level CLIP loss function, human interpretability can be effectively measured, ranging from shallow geometric to deep semantic perspectives. By optimizing both BCE loss and CLIP loss under different complexity constraints, the ISQA system can achieve a decent tradeoff in the triangular metric.

## 4 Emergent Communication System

### 4.1 System overview

Based on the task formulation and the evaluation in the previous section, this section proposes an emergent communication system to accomplish this task; see an overall illustration in Figure 2. The proposed system comprises a sender module and a receiver module. Both are trainable networks to realize the roles of the sender and receiver in this task. We now elaborate the details of each module.

### 4.2 Sender Module

Sender module is designed to improve the drawing ability given specific requirements, including the drawing complexity and human-interpretability, so that the receiver can easily capture information.

**Overall procedure.** Here we consider a drawing pipeline with a varying complexity. Let $\mathbf{X}$ be the input RGB image, $\mathbf{H}_i$ be the feedback sketch in the $i$th round, $a$ be the human interpretability level that determines the coefficient of $L_2$ in Eq 3 and $b_i$ is the drawing complexity of the sketches. In the $i$th round, the sketch $\mathbf{S}_i$ generated by the sender can be obtained as,

$$\mathbf{F}_{\text{geo}} = f_{\text{geo}}\left(\mathbf{X}\right), \ \mathbf{F}_{\text{prag}} = f_{\text{prag}}(\mathbf{X}), \tag{4a}$$

$$\mathbf{F}_{\text{fusion}} = f_{\text{fusion}}(\mathbf{F}_{\text{geo}}, \mathbf{F}_{\text{prag}}, a), \tag{4b}$$

$$\widehat{\mathbf{S}} = f_{\text{sketch}}\left(\mathbf{F}_{\text{fusion}}, \sum_{\tau=1}^{i} b_\tau\right), \tag{4c}$$

$$\mathbf{S}_i = f_{\text{select}}(\widehat{\mathbf{S}}, \mathbf{H}_i, b_i), \tag{4d}$$

where $\mathbf{F}_{\text{geo}}, \mathbf{F}_{\text{prag}}, \mathbf{F}_{\text{fusion}}$, are feature maps, $\widehat{\mathbf{S}}$ represents the sketch before undergoing the selection procedure to obtain the hard complexity constraints.

**Image encoding.** In Step (4a), we consider two encoders: a geometric encoder, which extracts geometric features for better sketching, and a pragmatic encoder, which extracts pragmatic features for better semantic information. The geometric encoder $f_{\text{geo}}(\cdot)$ encodes an RGB image into high-dimensional representations. The geometric encoder $f_{\text{geo}}(\cdot)$ is a pretrained model with fixed parameters introduced in [43] with a generative adversarial network. This encoder is designed to generate sketches with high geometrical similarity and can extract features that are closely relevant to the process of sketch generation. In addition, another pragmatic encoder, $f_{\text{prag}}(\cdot)$, which consists of several ResNet[45] blocks, is trained to minimize loss $L$ under the game setting. The inclusion of a trainable pragmatic encoder in the proposed model provides the sender with the ability to encode the input image into highly compressed features that are suited to situations where the complexity constraint is set to be extremely low.

**Feature fusion.** In Step (4b), the feature fusion module $f_{\text{fusion}}(\cdot)$ is used to fuse geometric features and pragmatic features based on a given different human interpretability level. To implement, $f_{\text{fusion}}(\cdot)$, we consider a simple convex combination of geometric and pragmatic features:

$$\mathbf{F}_{\text{fusion}} = \text{Deconv}\left(a\mathbf{F}_{\text{geo}} + (1-a)\mathbf{F}_{\text{prag}}\right), \tag{5}$$

where $a$ is the hyper-parameter that controls the relative weight of the $L_1$, $L_2$ losses and Deconv () is a deconvlutional network. When $a = 1$, the objective is to minimize the CLIP distance between the generated sketch and the original image, thereby maximizing human interpretability. Conversely, when $a = 0$, the objective is to maximize the SQA performance without considering human interpretability. By tuning the hyper-parameter $a$, the sender can adjust the weight balance between human interpretability and downstream SQA task performance.

**Sketch generation.** Step (4c) generates the sketch from the fused feature with the drawing complexity constraints. Initially, the drawing complexity $b_i$ is transformed into an indication vector, which is further encoded using a multi-layer perceptron to produce the drawing-complexity matrix. The drawing-complexity matrix is then concatenated to $\mathbf{F}_{\text{fusion}}$, yielding a complexity-aware feature map. This feature map is then fed to the decoder, which is composed of a U-Net[46] and multiple residual blocks to decode the complexity-aware feature map into the reconstructed sketch $\widehat{\mathbf{S}} \in \mathcal{R}^{H \times W}$.

**Pixel Selection.** To ensure that the communication consumption does not exceed the upper bound, all pixels are ranked according to their value, and the top $b_i N$ pixels are selected to display in the final

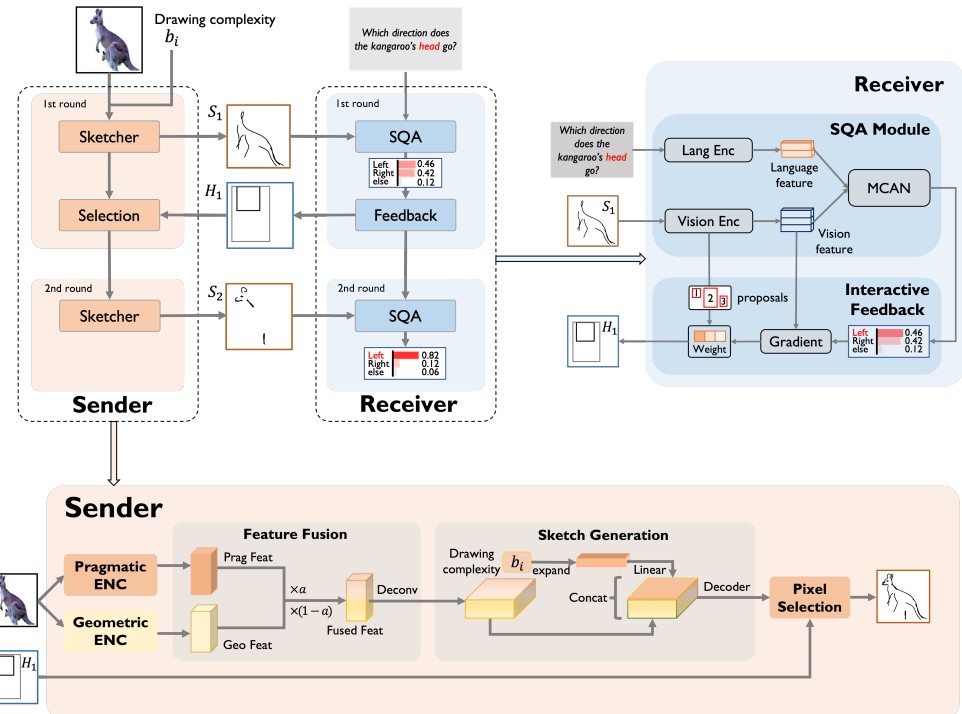

Figure 2: System overview with two interaction rounds.

sketch, while the other pixels are discarded, where $N$ denotes the number of pixels in original RGB image. By combining soft complexity encoding and hard selection procedures, we build a network that is guided by complexity constraints to generate suitable sketches for different situations, while the hard selection ensures that the complexity will not be exceeded. The selection mechanism can be naturally extended to select crucial pixels in multi-round settings. After receiving $\mathbf{H}_i$, the sender rerank the importance of each pixel according to their value times corresponding weight in $\mathbf{H}_i$. After removing previously transmitted pixels, the sender adds the top n remaining pixels to the sketch in subsequent rounds.

### 4.3 Receiver Module

The duty of the receiver module is to answer the question according to the sender module's sketch and provide feedback. The key of the receiver module is to associate the question with the critical spatial areas in the sketch. Our receiver module consists of two parts: SQA and interactive feedback mechanism. SQA is a modified VQA module that takes questions and sketches as input and select an answer from all the options. Interactive feedback module consists of a reasoning module that couples tightly with the detector-based vision encoder and a selection mechanism.

**SQA module.** The SQA module is primarily based on MCAN [35], with the addition of our pretrained sketch Faster-RCNN. Let $\mathbf{Q}$ be the question, $\mathbf{S}_i$ be the sketch, and $\widehat{\mathbf{A}}_i$ be the predicted answer distribution in the $i$th round. The SQA module works as follow,

$$\mathbf{F}_{\text{language}} = f_{\text{language}}(\mathbf{Q}), \tag{6a}$$

$$\mathbf{P}_i, \mathbf{F}_{\text{vision},i} = f_{\text{vision}}(\sum_{\tau=1}^{i} \mathbf{S}_\tau), \tag{6b}$$

$$\widehat{\mathbf{A}}_i = f_{\text{coattn}}(\mathbf{F}_{\text{language}}, \mathbf{F}_{\text{vision},i}), \tag{6c}$$

where $\mathbf{P}_i$, $\mathbf{F}_{\text{vision},i}$ and $\widehat{\mathbf{A}}_i$ denote the proposals, vision feature and answer distribution in the $i$-th round, respectively, and $\mathbf{F}_{\text{language}}$ is the language feature. In Step (6a), questions are tokenized into words and transformed into vectors using pre-trained word embeddings $f_{\text{language}}(\cdot)$ [47]. In Step (6b), a Faster-RCNN-based [48] visual encoder $f_{\text{vision}}(\cdot)$ extracts features from the accumulated sketch and

return corresponding proposals $\mathbf{P}_i$. To train this sketch encoder, we create the sketch-VG dataset[49] by utilizing [43] and pretrain the vision encoder on that to replace RGB-based decoder on MCAN [35]. The visual and language features are subsequently passed on to the modular co-attention layers in the MCAN architecture, $f_{\text{coattn}}(\cdot)$ in Step (6c) to calculate the $\widehat{\mathbf{A}}_i$, while the proposals $\mathbf{P}$ are forwarded to the interactive feedback module for further attention allocation.

**Interactive feedback module.** To facilitate efficient interactive feedback with human interpretability, it is natural to return a feedback sketch that indicates which parts of the sketch are more relevant to the question. To obtain this feedback sketch, our intuition is to analyze the relevance between the predicted answer distribution and each feature via the well-known Grad-CAM [50] method. In the $i$th round, the feedback module works as follow,

$$\beta_i^k = \sum_j \frac{\partial \sum_{v=1}^l \widehat{\mathbf{A}}^v}{\partial \mathbf{F}_{\text{vision},i}^{k,j}} \in \mathcal{R}, \tag{7a}$$

$$w_i^j = \text{ReLU}\left(\sum_k \beta_i^k \mathbf{F}_{\text{vision},i}^{k,j}\right) \in \mathcal{R}, \tag{7b}$$

$$\mathbf{H}_i^j = \frac{w_i^j}{E_i^j}\mathbf{P}_i^j \in \mathcal{R}^{H \times W}, \tag{7c}$$

$$\mathbf{H}_i = \sum_j^n \mathbf{H}_i^j \in \mathcal{R}^{H \times W}, \tag{7d}$$

where $\beta_i^k$ represents the weight of the feature map in the $k$th channel, $v$ denotes the index of top $l$ possible answer, $\widehat{\mathbf{A}}^v$ denotes the probability of answer index $v$, $\mathbf{F}_{\text{vision},i}^{k,j}$ refers to the vision feature of proposal $j$ in channel $k$, $E_i^j$ is the area of proposal $j$ and $\mathbf{P}_i^j$ is a mask matrix reflecting the spatial area of a proposal, where the element located in the $j$-th proposal is 1, and 0 elsewhere, $\mathbf{H}_i^j$ is a weighted mask matrix associated with proposal $j$ whose weight reflects its importance and $\mathbf{H}_i$ is the feedback sketch in the $i$th round.

In Step (7a), we calculate the weight $\beta^k$ of each channel by computing the partial derivative $\widehat{\mathbf{A}}$ with respect to $\mathbf{F}_{\text{vision}}$ following the original Grad-CAM procedure. In Step (7b), we utilize $\beta^k$ to sum up the channel dimension and get weight $w^j$ of each proposal $j$. Combining these two steps, we leverage the gradient flowing to assign importance value to each feature vector of a particular proposal. In Step (7c), the attention matrix for proposal $j$ is computed by dividing the weight of the proposal by its area, in order to avoid bias towards large proposals that may otherwise dominate the attention. In Step (7d), we aggregate the bounding boxes of each proposal to obtain the final feedback sketch $\mathbf{H}_i$. Notably, $\mathbf{H}_i$ can be simply represented as a bounding box and the corresponding weight, which only needs five numbers, significantly reducing the drawing complexity.

## 5 Experiments

### 5.1 Dataset

Visual Question Answering (VQA) v2.0 [36] is a dataset that containas open-ended questions about images that require an understanding of vision, language and commonsense knowledge to answer. To fit with our SQA game, we remove all questions that include color-related word since sketch has no information about color. This reduces the size of dataset by about 15%. Visual Genome (VG) [49] is a dataset that contains 108K images with densely annotated objects, attributes and relationships. Based on VG dataset, we generate the corresponding sketches by the sketching model [43] to train our Faster-RCNN model required by ISQA tasks. The evaluation metrics of VG datasets include the mean average precision(mAP) with IOU theshold 0.5(mAP@50) and the mAP@0.5 weighted by the number of boxes for each class(weighted mAP@50).

### 5.2 Experimental setting

With aforementioned optimazation object $L$, we trained three different models with SQA module pretrained under VQAv2 dataset and the images are transferred to sketches with [43]:

1. *Pragmatic*: trained to minimize $L_1$, equivalent to minimize $L$ with $a = 0$.
2. *Geometric*: trained to minimize $L_2$, equivalent to minimize $L$ with $a = 1$.
3. *PraGeo*: trained to minimize $L$ with $a = 0.5$.

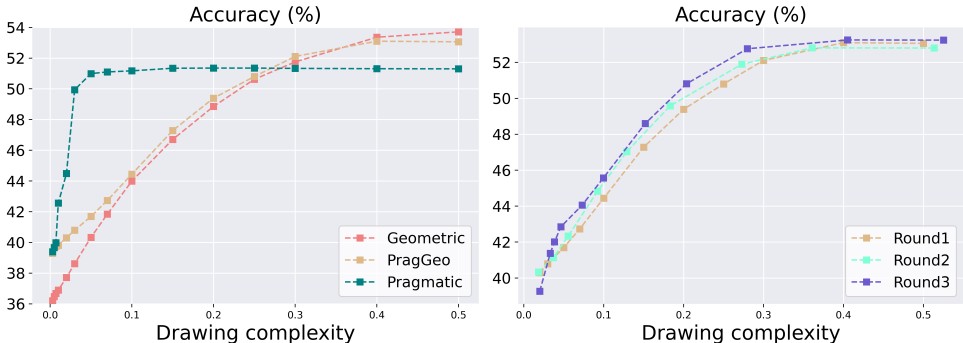

Figure 3: Comparison of *Pragmatic*, *Geometric*, *Prageo*(left) and *Prageo* models with different rounds(right). The $x$-axis is $\sum b_i$.

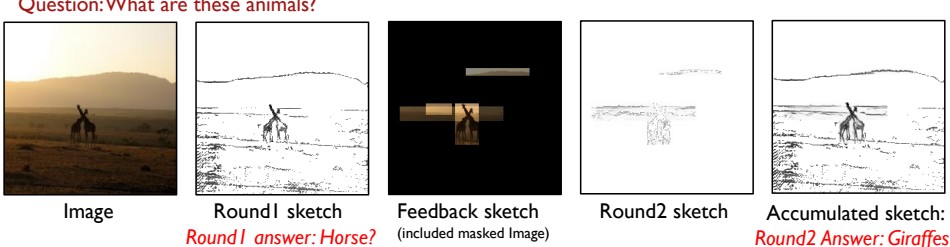

Figure 4: Multi-round interactive SQA. From left to right we display RGB image, sketch transmitted in round 1, RGB image masked with $\mathbf{H_i}$, extra pixels added in round 2, and the whole sketches in round 2.

## 5.3 Quantitative evaluation

Fig 3 compares the ISQA accuracies of *Pragmatic*, *Geometric* and *PraGeo* models as a function of the drawing complexity $B$ , where $x$-axis is $\sum b_i$ and $y$-axis is the answer accuracy. For *PraGeo*, we consider both one-round and multi-round settings.

**Comparison among *Pragmatic*, *PraGeo* and *Geometric* models.** 1) When $B$ is set to be lower than $0.3N$, *Pragmatic* outperforms other methods in overall accuracy and *PraGeo* outperforms *Geometric* in complexity situations. 2) When $B$ is larger than $0.3N$, the accuracy of geometric model catches up with other methods'. 3) The peak accuracy of pragmatic model is lower than other models' since *Pragmatic* is easier to overfit in low-complexity situations.

**Effectiveness of interactive feedback mechanism.** 1) Multi-round interaction can improve the SQA accuracy. 2) The accuracy gap between one-round *PraGeo* and multi-round *PraGeo* are more significant when complexity $B \in (0.05N, 0.3N)$ since: when the complexity is too low reasoning module is unable to infer sufficient useful information and interaction cost extra bandwidth consumption; when complexity is appropriate, the bandwidth expended on this extra interaction is relatively negligible and reasoning module generate effective feedback according to the round 1 sketches; when the complexity is too high, most of the crucial information is transmitted in round 1. 3) When comparing two-round interaction to three-round interaction, we see that the difference is minor when compared to the gap observed between one-round and two-round interaction.

**Human interpretability evaluation.** Table 1 compares the human interpretability of *Pragmatic*, *Geometric* and *PraGeo* by calculating CLIP distance. We see that i) without any constraint in training, *Pragmatic* has a poor human interpretability, and ii) *PraGeo* is much closer to *Geometric*.

|  | Pragmatic | PragGeo | Geometric |
|---|---|---|---|
| Human interpretability Score | $0.998 \pm 0.002$ | $0.600 \pm 0.068$ | $0.478 \pm 0.108$ |
| CLIP Distance | 0.9077 | 0.1815 | 0.1193 |

Table 1: Average CLIP distances and human evaluation of three models.

To verify the consistency between CLIP and manual measurement, we conduct a human evaluation, where 12 participants were tasked with assessing the human-interpretability of three different models. Participants were presented with three corresponding images from each model simultaneously and asked to score them based on judgment criteria ranging from 0 to 1. (lower is better, 0, 1 represent

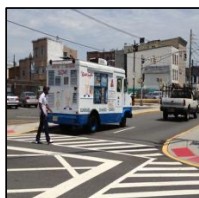 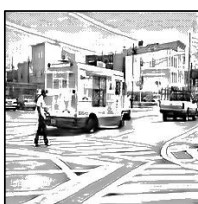 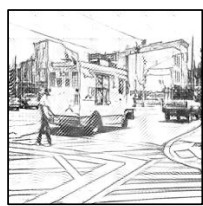 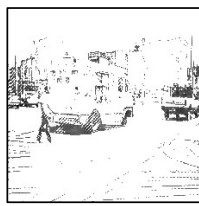 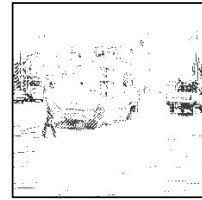

Figure 5: Sketches under multiple drawing complexities. From left to right, there are RGB-images, sketch without complexity constraint, and sketch generated with *PraGeo* model, when $B = 0.5N, 0.3N$ and $0.1N$

fully understand and can not understand, respectively). We sampled 3,000 sets of images and each set comprised images generated by the Pragmatic, PragGeo, and Geometric models from the same RGB images. We see that: 1) the results in human experiments and the CLIP distance are consistent to show the order of Geometric, PragGeo Pragmatic in terms of human interpretability; 2) both PragGeo and Geometric models provide an human interpretable sketches while Pragmatic models can not be understood; and 3) Comprehensively considering with Fig.3, the PragGeo achieves a more optimal balance among the task performance and human interpretability.

Besides, to further verify the human interpretability, we sampled 300 images and questions pairs from the dataset and test human substituting as the receiver. Ten people are involved in this experiment and table. 2 shows the average accuracy and variance of people and machine receiver. We see that human achieve better accuracy in number question but failed to outperform machine in Yes-or-No and other question. The experiments show that our architecture can deliver a decent human interpretability and our sender can work with human.

|  | Yes-or-No | Number | Other |
|---|---|---|---|
| Human | $66.22 \pm 4$ | $39.71 \pm 4.33$ | $28.57 \pm 11.30$ |
| Machine | 70 | 26 | 38 |

Table 2: Results of human substituting as the receiver.

## 5.4 Qualitative evaluation

**Multi-round interaction.** Fig. 4 illustrates the procedure of multi-round interaction. We see: i) the receiver generates the feedback sketch which contains crucial region according to the preliminary assessment; ii) the sender provides extra pixels when round is more than 1 according to the feedback sketch; and iii) the receiver modifies the answer in accordance with the updated accumulated sketch.

**Drawing complexity.** Fig. 5 presents the sketches generated by *PraGeo* model under various drawing complexities. We see that our *PraGeo* model can preserve human interpretability when the drawing complexity is sufficiently large ($B \geq 0.1N$).

**Human interpretability comparison.** Fig. 6 shows examples of sketches generated by the three models when $B = 0.03N$. We see that i) the sketches generated by the *Pragmatic* model are difficult for humans to understand; ii) *PraGeo* model can still capture the outlines of the bowls and chopsticks and outperform *Pragmatic* model.

**Interactive feedback mechanism.** Fig. 7 illustrates that interactive feedback mechanism generates $\mathbf{H}$ according to specific questions. The first row shows the RGB image and corresponding sketch, from left to right. When the question is "What are the people in the background doing?", the interactive feedback module generates $\mathbf{H}$ that specifically focuses on people to request more information about the area that is likely to contain humans. However, when the question is "What is he on top of?", the interactive feedback module choose to request more information about the area under the man.

**Insights brought by ISQA task.** First, information disparity is the prerequisite for interaction. Fig. 7 shows the feedback messages for the same image according two different questions. We see that our feedback message transfers question-related information to the sender, enhancing communication efficiency via a gradient-informed region of interest (can be displayed as sketch as shown in Figure 2, which enables a human-like multi-round interaction for the first time in visual emergent communication. This provides an insight that one of the reasons why interaction emerges might be querying task-aligned information when receiver is more acquainted with the object than sender. Second, the constrain of communication complexity promotes interaction. Fig. 3 shows the ISQA performance comparison between multi-round and one-round communication. We see that the

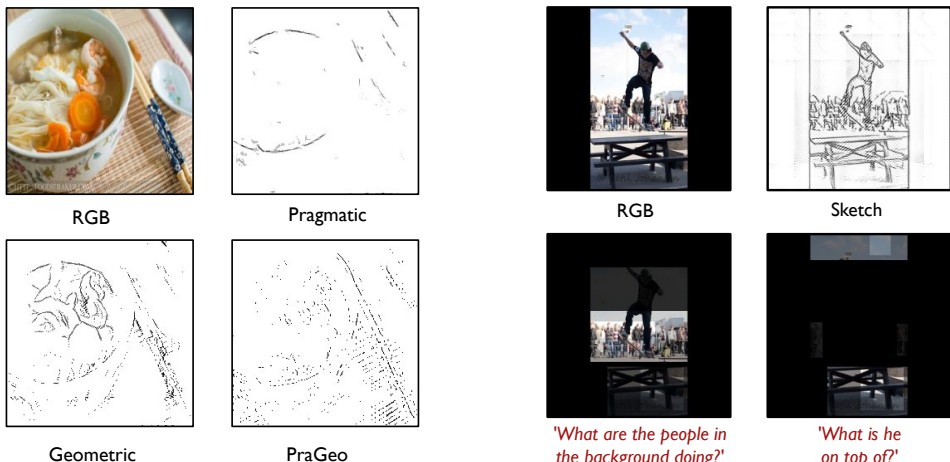

Figure 6: Sketches generated with different models when $B = 0.03N$.

Figure 7: For the same image, different questions lead to distinct feedback sketches.

downstream performance can be boosted when interaction is introduced to the visual communication pipeline. This provides an insight that the complexity constrains promote the agents to pursue a more efficient communication leads to interaction. Third, it requires sufficient input information for the feedback to be functional. Fig. 3 shows that functional feedback from the receiver relies on adequate communication complexity. This offers the crucial realization that precise feedback hinges on a foundational shared understanding.

## 5.5 ISQA study

Due to the differences in input modality between SQA and VQA, the RGB-based model is not suitable for SQA. Two key components, Faster-RCNN vision encoder and MCAN question answering module, thus need to be trained with a sketch-based dataset. Table 3 evaluates two Faster-RCNN models. We see: i) the RGB-based model is not applicable to sketch-based dataset. ii) sketch-based detector works normally on sketches but not as well as RGB-based model on RGB images.

Table 4 evaluates the performance of two MCAN models. We see that i) the RGB-based model suffers significant performance degradation on sketch-based dataset. ii) the performance of sketch-based MCAN is acceptable while 10.16 percent lower than RGB-based model in RGB-based datset, which also serves as the upper-bound of ISQA task.

| model | dataset | mAP@0.5 | Weighted mAP@0.5 |
|---|---|---|---|
| RGB-based | RGB | 0.082 | 0.149 |
| RGB-based | Sketch | 0.003 | 0.010 |
| Sketch-based | Sketch | 0.034 | 0.085 |

Table 3: Performance of Fast-RCNN-based models with ResNet50 backbone on VG and sketch-VG. The RGB-based model is trained on color images and the Sketch-based model is trained on sketches.

| model | dataset | overall | other | yes-or-no | numbers |
|---|---|---|---|---|---|
| RGB-based | RGB | 64.74 | 52.44 | 83.41 | 47.28 |
| RGB-based | Sketch | 40.86 | 20.88 | 67.48 | 23.19 |
| Sketch-based | Sketch | 54.58 | 40.81 | 74.81 | 37.01 |

Table 4: Accuracy of RGB-based MCAN and sketch-based MCAN on RGB/sketch-based VQAv2 dataset. The RGB-based model is trained on color images and the Sketch-based model is trained on sketches.

## 6  Conclusion

This study proposes a new interactive sketch question-answering task along with an interactive emergent communication (EC) system to facilitate bidirectional communication between two players. Our approach employs an interactive feedback mechanism that allows the EC system to provide feedbacks to guide sketch generation. Our experiments reveal that the proposed multi-round EC system can achieve an effective balance among question answering ability, drawing complexity and human interpretability.

**Limitation and future work.** The current interactive EC system is limited to two collaborative players. In the future, we are going to explore interactive emergent communication among a group of intelligent agents, which is more complex and tricky on the design of task and interactive process. Collective intelligence has shown its capacity in applications such as collaborative perception, UAV cluster technology, social network analysis, etc. We believe that multi-agent interaction is a promising direction for artificial intelligence research in the future.

**Acknowledgement.** This research is supported by NSFC under Grant 62171276 and the Science and Technology Commission of Shanghai Municipal under Grant 21511100900, 22511106101, and 22DZ2229005

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
