# OpenReview forum: "Emergent Communication in Interactive Sketch Question Answering"
_NeurIPS.cc/2023/Conference — NeurIPS 2023 poster_

### Official Review · Reviewer_NLdK · 2023-07-05

**Soundness:** 3 good
**Presentation:** 3 good
**Contribution:** 3 good
**Rating:** 7
**Confidence:** 4

**Summary:**

The work focuses on multi-turn sketch-based emergent communication.
Authors propose a novel two-round interactive task, named Interactive Sketch Question Answering (ISQA).
They suggest an architecture and an implementation, based mainly on existing components (MCAN, Fast-RCNN) while incorporating several novel ideas such as 1) dynamically restricting the channel capacity by controlling the number of transmitted pixels, and 2) providing feedback from receiver to sender via focus boxes.
They suggest a triangular evaluation method that seeks a balance between human interoperability and task accuracy.

**Strengths:**

The main strength of this paper is by suggesting a two-turn visual communication game that nicely models the need for two parties to communicate, with partial observability, to solve a task.
In addition, the paper demonstrates a method to achieve a nice balance between interoperability and pragmatism.
The most interesting observation, to my mind, is provided in lines 304-306 where the authors show that when the complexity is too low, the reasoning module cannot infer sufficient useful information in a single round and thus needs to request a more focused information (a clarification question).
The way the architecture is composed and implemented for modeling the problem at hand is not trivial and interesting.


**Weaknesses:**

The authors assess human interpretability using the CLIP model. Doing an actual human survey of the results would be more appropriate.

The interoperability/pragmatism balance is essentially solved by adding a CLIP-based loss that provides additional supervision towards human interpretability, which is not aligned with the intention to model communication emergence.

Experimental datasets are not described in enough detail. For example, it is unclear how the three reported tasks (Yes-No, Number, Other) correspond to the two described datasets.

Results are not totally consistent (for example, in the Yes-No task where the PraGeo is lower than both the geometric and pragmatic models) and more experiments over more datasets seems needed.

Notations and explanations can be further worked out to assist the reader. See some examples in the Questions section.

Maybe worth mentioning references:
Pragmatic inference and visual abstraction enable contextual flexibility during visual communication, by Judith E. Fana, Robert X.D. Hawkins, Mike Wub and Noah D. Goodman,


**Questions:**

In section 3.1 what are the dimensions of H_i and A_i? (explained later)
Line 150 - will be good to stress the fact that b_i is a ratio (explained later)
Line 244 – will be good to explain what proposals are.
Lines 293-297 the x-axis is not easily defined (line 293) and then referred to as 0.1N, 0.3N, which are hard to find in the graphs. Can’t you use the 0.xN scale? or at least add the values you refer to as labels to the x-axis?
Datasets are missing the explanation of complexity/difficulty of tasks, namely yes/no, number and other which you refer to in the figure. A baseline random accuracy can also be helpful to add or mention.
Table-1: is the lower the better? Worth mentioning.

**Limitations:**

limitation section is provided.

---

> ### Author Rebuttal · Authors · 2023-08-10
>
> Thank you for your time and valuable suggestions. Here are our detailed responses.
>
> ---
>
> **W1: Human survey**
>
> We agree with your opinion and added a human survey. The human evaluation results are consistent with the CLIP distance. See general response 3.
>
> ---
>
> **W2:  CLIP-based loss provides additional supervision.**
>
> We think leverage CLIP model is reasonable for four reasons.
> - The primary rationale behind striving for human interpretability is to achieve a consensus with humans on various concepts. This ensures that the emergent communication resonates with human understanding and can be intuitively comprehended.
> - We view the CLIP model as an exceptional tool for learning human consensus and perceptual abilities. It serves as a teacher for our systems to better align with human cognitive results.
> - Previous methods, such as [13], have also leveraged VGG classification models during training. This precedent signifies the utility and relevance of employing such models to drive desired outcomes.
> - While the idea of entirely free emergence is indeed exhilarating, we recognize that there's a gap between this ideal and current research paradigms. Our approach is an endeavor to traverse this gap, balancing the excitement of pure emergence with the practicality of human-aligned understandings.
> We believe that, through these methods, we maintain the spirit of modeling communication emergence while simultaneously ensuring that the results align with human interpretability."
>
> ---
>
> **W3 & Q5: Details about experimental datasets.**
>
> Yes-or-No, Number and Other are different question types in the dataset(according to the answer).
>   - Yes-or-No means the answer to the question is yes or no. This type of question is easier to solve and mainly focuses on a single object or pattern in the image.
>   - Number means the answer is Arabic numerals and often requires detailed information.
>   - Other includes remaining questions, whose answers include more than 3000 alternative words or phrases.
>   - We will add the related explainations to the updated paper.
>
> ---
>
> **W4: Result consistency.**
>
> We confirm that the results are consistent and reliable. The observation mentioned by the reviewer is commonly known as Simpson's Paradox, where overall statistical results are inconsistent with the local statistical results. Yes-or-No questions often require less information and have a higher possibility to overfit and strong randomness since it has only two options.
>
> ---
>
> **W5: Further notations and explanations.**
>
> Thanks for your advice! We will correct those issues in the revised version.
>
> ---
>
> **W6: References.**
>
> Thanks for your advice! We will add those references and do a more comprehensive literature review in the recognitive community.
>
> ---
>
> **Q1**:
>
> The dimension of $H_i$ is $m\times 4$ and the dimension of $A_i$ is $n\times 1$. Where $m$ denotes the number of regions and $n$ denotes the number of alternative answers. 4 means $(x, y, height, width)$ to locate a box in the sketch.
>
> ---
>
> **Q2**:
>
> Thanks for your advice, we will modify our updated paper.
>
> ---
>
> **Q3**:
>
> The "proposals" are thus a critical part of the Faster R-CNN architecture, as they define the potential regions where objects might be present. Here's how proposals are generated in more detail:
>   a. Anchors: At each position of the image, the RPN proposes multiple regions of different scales and aspect ratios. These are called anchors. For example, you might use 3 scales and 3 aspect ratios, for a total of 9 anchors at each position.
>   b. Classification and Regression: For each of these anchors, the RPN predicts two things:
>     - Whether it contains an object or not (classification).
>     - Adjustments to the anchor box to better fit the object (regression). These adjustments are typically four real-valued numbers for the coordinates of the top-left and bottom-right corners of the box.
>   c. Proposal Selection: From all the proposed regions, a set number (e.g., 2000) of the best scoring regions are selected as region proposals using a technique called Non-Maximum Suppression (NMS) which helps in eliminating multiple detections for the same object.
>
> ---
>
> **Q4**:
>
> Sorry for inconvenience, we will modify the x-axis in the updated version.
>
> ---
>
> **Q5**:
>
> Please see W3.
>
> ---
>
> **Q6**:
>
> Thanks for your advice! The lower is the better for the CLIP distance. Besides we will add human survey. See general response 4.
>
> ---
>
> Overall, we hope that our responses can fully address your concerns. We will be grateful for your further feedback.

---

> > ### Comment · Reviewer_NLdK · 2023-08-14
> > **I read and acknowledge authors responses.**
> >
> > Thanks and good luck.

---

### Official Review · Reviewer_jxDd · 2023-07-06

**Soundness:** 3 good
**Presentation:** 3 good
**Contribution:** 3 good
**Rating:** 6
**Confidence:** 4

**Summary:**

This paper proposed a new multi-round visual communication task with an interactive system for emergent communication. During the game, the sender needs to sketch on the canvas to communicate a target image, while the receiver needs to answer a question regarding the target image and give feedback on the sender’s sketch. The training framework balances task accuracy, drawing complexity, and human interpretability. Experimental results show that the agents can communicate successfully using sketches and feedback. The emerged sketches can maintain interpretability and high accuracy with lower complexity. And the feedback given by the receiver can effectively enhance the performance.


**Strengths:**

1. This paper proposed a novel setting where each of the agents can only observe a partial environment that necessitates the feedback of the receiver. And the feedback is smartly provided in a sender-understandable way (bounding boxes). Compared with the previous work, this environment enables bi-directional communication where both agents can “draw” on the canvas.
2. The training framework considers triangle optimization – task accuracy, drawing complexity, and human interpretability.


**Weaknesses:**

1. Complexity B:
for the complexity in section 5.3, what is the specification for $b_i$ and $h_i$ separately?
It will be interesting to know how which agents contribute more to the efficiency – while achieving high accuracy, is the high efficiency due to the sender drawing less or the receiver giving more accurate feedback?
2. The maximum round: only models trained with two-round are reported. Is there a reason that the maximum round is set to 2? Given more rounds, the performance change can help us understand whether one round of feedback from the receiver is enough to finish the task.


**Questions:**

Why is the $\sum b_\tau$ given to the sketch model? Would $b_i$ be sufficient?


**Limitations:**

It will be interesting if the agents can control the complexity of the sketch based on the target image and the receiver’s feedback. Similarly for the number of bounding boxes at the receiver’s side.

---

> ### Author Rebuttal · Authors · 2023-08-10
>
>
> Thank you for your time and valuable suggestions. Here are our detailed responses.
>
> ---
>
> **W1:** Which agents contribute more.
>
> Our setting considers a collaboration between the sender and the receiver. Through the proposed feedback mechanism, both agents have equal rights to affect the interaction and either agent cannot be separately considered. The reason why sender can use less complexity to transmit sufficient information is because the receiver gives it an accurate feedback while the accurate feedback constructs on the decent round1 communication. To illustrate, this can be likened to a teacher identifying key contents for students prior to the final exam. By concentrating efforts within key contents, students are able to achieve commendable results with less time. Meanwhile, the reason why the teacher accurately point out these key contents is because the teacher communicate sufficiently with the students, thereby gaining understanding into students strength and weakness of the curriculum.
>
> Overall, the both components collaborate to deliver a superior performance.
>
> ---
>
> **W2:** Maximum round.
>
>  2-round interaction is enough to finish the task. The 3-round model provides slightly lower performance compared to 2-round model. There are two reasons why 3-round can not improve performance: First, a 3-round approach will reduce the complexity budget assigned to the first and will reduce the accuracy of the feedback module(See line 304-306 for reasons). Second, each round of interaction requires an extra complexity for the feedback message, causing the complexity assigned to drawing decreases.
>
> ---
>
> **Limitation:**
>  Thanks for your advice! We will explore how to impel the sender and receiver to control complexity in future work.
>
> ---
>
> Overall, we hope that our responses can fully address your concerns. We will be grateful for your further feedback.

---

> > ### Comment · Reviewer_jxDd · 2023-08-19
> >
> > Thanks for the detailed rebuttal. It would be better that the clarification or further experiments on the maximum round can be added to the revision. I will keep my rating unchanged and recommend accepting this paper.

---

> > > ### Author Response · Authors · 2023-08-20
> > > **Thanks for your reply!**
> > >
> > > Thanks for your kindly reply! Since the further experiments on the maximum rounds takes a lot of time and computation resources, we will add them to the updated version of the paper.

---

### Official Review · Reviewer_APpT · 2023-07-07

**Soundness:** 1 poor
**Presentation:** 3 good
**Contribution:** 1 poor
**Rating:** 4
**Confidence:** 3

**Summary:**

In this paper, the authors proposed a new task about emergent communication by tackling visual question answering as an iterative sketch question answering process. The authors proposed a three-factor evaluation metric, including question answering performance, drawing complexity and human-interpretability. A framework consisting of Sender and Receiver is proposed to perform multi-round interaction to tackle the proposed task. VQAv2 is used for empirical evaluation of the proposed framework.

**Strengths:**

1. The problem setting is very interesting.
2. The proposed method is intuitive and straightforward.
3. The paper is presented clearly and easy to follow.

**Weaknesses:**

1. The new insight is limited.

a. Visual question answering is indeed a new task compared to classification. But what is unique in terms of emergent communication when visual question answering is used as the target task? From the current manuscript, there is not really a metric and any empirical evidence showing the improve communication quality over [14].

b. Despite the authors target at multi-round interaction, the two settings evaluated are one-round and two-round.From the visualizations, the sketch used is usually the sketch of the main object. There doesn't seem to be a pattern in terms of communication with only one/two round of communication.

c. More fundamentally, how does communication emerge and how does communication gets better/more efficient when the task is more complex? The reviewer feels these fundamental questions still left unsolved and the current manuscript didn't show any potential of helping solve these problems.

2. Empirically, current evaluation is not sufficient enough. Currently, the communication quality is mainly measured through automatic metric like CLIP-based score. There should be some quantitative analysis verifying the correlation between automatic score and manual measurement.

**Questions:**

Please check the weakness for details.

**Limitations:**

Need more discussion on the fundamental questions of emergent communication.

---

> ### Author Rebuttal · Authors · 2023-08-10
>
> # Response to Reviewer APpT
>
> Thank you for your time and valuable suggestions. Here are our detailed responses.
>
> ---
>
> **W1:**  a) Uniqueness of the proposed task? Comparison to classification. Comparison with [14]. b) Pattern with one/two round communication.  c) Communication gets better when the task is more complex?
>
> **a)** We address this point from three aspects:
>   1) The uniqueness of the proposed visual question answering task for emergent communication are three-folds: information disparity, feedback mechanism and  triangle evaluation; see more details in General Response 1.
>   2) The proposed visual question answering task setting is better at promoting emergent communication than the classification task, because it can deliver a higher resemblance to human communication in the real world. Human interactions go far beyond merely classifying images; more often, they involve bilateral communication to achieve a complicated task. From this perspective, our question-answering task is more closely aligned with daily human interactions than the common Lewis' game in previous works.
>   3) It is hard for us to experimentally compare with [14], because the aims and task settings are different and incomparable; see more details in General Response 3.
>
> **b)** There is one distinct pattern in 2-round communication: for the same image, each question lead to a unique sketch in the second round, reflecting the interaction successfully helps the sender focus on question-related content. As depicted in Figure 7, it is evident that for disparate questions, the weight assigned to different regions varies considerably, leading to a prioritization of pixel transfer from lighter areas. For question "“How many people can you see in the picture?", the sketch transferred in the second round will focus on the people while for question "Is it still snowing in the picture?", the sketch transferred in the third round will be presented in a more global fashion. The fact shows that the sketch in the second round depends on the question and definitely will have different patterns when compare with round 1.
>
> **c)** Through making the task more complex, this work shows a lot of potentials to promote emergent communication. From an intuitive aspect, it is natural for a more complex task to promote emergent communication, because there is no need to create a sophisticated linguistic system if the task for collaboration is very simple. This work provides a recipe for designing a complex task through creating  information disparity, designing feedback mechanism and proposing triangle evaluation. From an experimental aspect, we have found a series of interesting patterns; see more details in General Response 2.
>
> ---
>
> **W2:**  Verify the correlation between automatic score and manual measurement.
>
> To verify the correlation between CLIP and manual measurement, we conduct a human evaluation, where 12 participants were tasked with assessing the human-interpretability of three different models. Participants were presented with three corresponding images from each model simultaneously and asked to score them based on judgment criteria ranging from 1 to 5 (higher score indicating better interpretability); see Table.1 in rebuttal PDF. We sampled 3,000 sets of images consistent with the settings in Table 1 of the paper. Each set comprised images generated by the Pragmatic, PragGeo, and Geometric models from the same RGB images.
>
> Table.2 in rebuttal PDF compares human interpretability score and CLIP distance. We see that: 1) the results in human experiments and the CLIP distance are consistent to show the order of Geometric, PragGeo Pragmatic in terms of human interpretability; 2) both PragGeo and Geometric models provide an average human interpretability score between partially understand and mainly understand while the sketches provided by Pragmatic models can not be understood; and 3) Comprehensively considering with Fig.3, the PragGeo achieves a more optimal balance among the task performance, drawing complexity and human interpretability.
>
> ---
>
> Overall, we hope that our responses can fully address your concerns. We will be grateful for your further feedback.

---

> > ### Comment · Reviewer_APpT · 2023-08-16
> > **Concerns remain**
> >
> > Thanks for the rebuttal.
> >
> > 1. The reviewer understands that the proposed model is not directly comparable with [14]. This doesn't necessarily mean that comparing with other tasks is not feasible. For example, the authors could build a classification version of the dataset by only using the labels of object categories of the COCO dataset where VQA is built upon.  Further degraded version of the tasks could be built by choosing different ways of communication (sketch or binary). Based on this, apple-to-apple comparisons can then be done.
> >
> > 2. The reviewer still thinks two rounds are not enough to really show the complexity of emergent communication. Changing the focus the sketch based on the feedback doesn't necessarily reflect the complexity of communication. The reviewer thinks there is some relationships between the communication bandwidth (B) and the number of rounds towards accomplishing the task. The current evaluation really is too limited in terms of one important aspect of communication: communication is done by modeling a sequence of signals.
> >
> > 3. The reviewer also has some follow-up questions on the human evaluation. How is the evaluation done exactly? Is there any training process for the human evaluators? How is each criteria defined?
> >
> > 4. Can the authors comment on fundamental questions of emergent communication and which are directly answered in this paper?

---

> > > ### Author Response · Authors · 2023-08-18
> > > **Thanks for your reply**
> > >
> > > 1. We appreciate the reviewer's wonderful suggestions about the comparison experiments. We agree that a fair comparison with the iconic previous work [14] would be great for the community. We aim for our work to serve as an expansion and exploration of an informative feedback of emergent communication, thus allowing a more comprehensive consideration of emergent communication together with [14]. Therefore,
> > > - For ISQA task with the binary flag, we will add experiments immediately.
> > > - For the degraded version of the task, it is hard for us to finish the model training and evaluation before Aug. 21. But we will add those experiments to the updated version of paper.
> > > ---
> > > 2. If we understand the reviewer right, the reviewer means that more-round communication, a better setting to promote emergent communication. We agree with the reviewer and this is exactly why we promote multi-round interaction and propose the feedback mechanism.
> > > - Our method supports more than 2 round communications. We are working on the experiments for 3 and 4 rounds. Since it will take about one week for the train and evaluation, we will add the experimental results to the updated version of our paper.
> > > - In this work, we set an upper bound for the number of communication rounds. Like in Avalon game, there are at most 5 tasks. The reason we set an upper bound is to make the setting friendly for time and computation. In ISQA, some questions with more than 3000 options, are very challenging even for humans. Continuously gaming without limits on communication rounds, can be both time-intensive and computationally demanding. In [14], the maximum communication rounds is 7, which is impressive, but there remains a ceiling. Additionally, the task in [14] focuses on classification, which is less GPU-intensive than question-answering.
> > > - Both the number of rounds and the message content are important for emergent communication. In terms of feedback messages, our approach expands [14]. Instead of using a binary flag, we employ continuous feedback messages that offer more detailed information.
> > > ---
> > > 3.
> > > - Evaluation: Participants see images including the RGB image and images generated by our Pragmatic, PragGeo and Geometric models simultaneously. Then they provide a score for the three generated images according to our criteria and their intuition.
> > > - Training: We provide a thorough explanation of our criteria to all participants, guiding them to evaluate based on their natural human intuition. We provide basic training for a few trials with the rating software.
> > > - Criteria:
> > >   - Understand all the content as well as RGB images.
> > >   - Understand a major part of the content, while not as clear as the RGB image.
> > >   - Understand some part of the content.
> > >   - Understand only few part of the content or finding few common features in the raw image and the sketch.
> > >   - Totally not understand anything.
> > > ---
> > > 4. Three fundamental questions of emergent communication are directly answered:
> > > - **What is a prerequisite for interaction in emergent communication?**
> > > Information disparity is a prerequisite for interaction. Fig. 7 shows the feedback for the same image according two different questions. We see that our feedback transfers question-related information to the sender, enhancing communication efficiency via gradient-informed region of interest (can be displayed as sketch as shown in Figure 2), which enables a human-like multi-round interaction for the first time in visual emergent communication. This provides an insight that one of the reasons why interaction emerges might be querying task-aligned information when receiver is more acquainted with the object than sender.
> > > - **Can multi-round interaction (feedback) promote more efficient communication?**
> > > Yes, through multi-round interaction, tasks can be accomplished more effectively while using less communication resouces. Fig. 3 shows two facts: i) when the 2-round and 1-round models have similar performance when $B > 0.2N$; and 2) the 2-round model can achieve superior ISQA accuracies compared with the 1-round model when the same complexity and $B \in (0.01N,0.2N)$.  These observations suggest that without communication constraints, multi-round communication do not necessarily provide more information than a single round. However, multi-round communication can optimize the use of communication resources and lead to more efficient exchanges.
> > > - **What is a prerequisite for multi-round interaction (feedback) to be beneficial?**
> > > For feedback to be effective, the receiver needs sufficient input information. Fig. 3 shows that the 2-round model does not have an advantage against the 1-round model when complexity constraint is too low ($B < 0.01N$). We see that only the feedback based on a minimal complexity requirements can be beneficial. This emphasizes that effective feedback depends on having sufficient background knowledge and essential preliminary information, a principle that resonates with broader human societal values.

---

### Official Review · Reviewer_ATWv · 2023-07-08

**Soundness:** 3 good
**Presentation:** 3 good
**Contribution:** 2 fair
**Rating:** 4
**Confidence:** 4

**Summary:**

The authors present a new problem setup for sketch-based emergent communication, distinguishing itself from existing work primarily through communication taking place iteratively over multiple rounds.  The authors also argue that the reliance on downstream tasks for evaluations allows for communication protocols to develop which are not necessarily easily interpretable by humans, and thus fail to fulfill this important goal of EC research.  Evaluating the behavior of various design choices, the authors show that they can prioritize different aspects of the problem: performance, but also human interpretability and drawing complexity (automated metrics)






**Strengths:**

- The idea of multi-round EC is very exciting!  I would love to see research move in this direction and, given just the difficulty of agents who learn when to talk (and talk with resonable sparsity), I imagine there are plenty of interesting problems to solve in that space.
- On that topic, the ability for the sketch model to generate very different sketches using the same image (when the question demands it) is demonstrated here and is a perfect example of what I would expect as sort of a main contribution from an EC model in this space.
- The problem setup and task are novel

**Weaknesses:**

- I won't dwell on this too much since it's primarily a track issue, and the paper could be considered by other merits, but it is difficult for me to consider this emergent communication at all.  It does however mean that a lot of EC motivation cited here doesn't seem very relevant upon reaching the experimental design section and understanding the learning problem.

- While the authors argue that existing work relied too much on downstream tasks for evaluation, regardless of this point, evaluating downstream did serve an important purpose in that it helped demonstrate some potentially useful application of the learned protocol.  Here the protocal seems rather contrived.  Of course something like a referential game is also rather contrived and I concede that point.  However, I'm willing to accept contrived environments if what emerges in the language is itself interesting and gives us some insight on what sorts of less contrived environments we may consider in future work.

- The drawer is vastly simplified when compared to the existing visual referential game work (cited here).  The authors state, "Vision-based EC systems have been developed with the aim of imitating the pictographic systems that were used in early human communication", but a pictographic system tends to abstract important visual features, sometimes caricaturing them for the purpose of clarity in communication.  I'm not convinced that this drawer is an appropriate substitute for this process.  If I want a giraffe drawn in 3 strokes vs. 8 strokes, we see the important visual features that are most characteristic of the giraffe.  If we are essentially revealing areas of an edge/depth-detected version of a real image, it seems very different.  From the examples of the sketches produced by various modes (pragmatic, geometric, prageo), none strike me as very similar in creating some simplified version of the original high-res image of the object, and a case should be made why this process could be considered an imitation of those systems in early human communication.

- While generating very different images from the same image when the question differs

- Is the binary flag model of [14] really that different from what occurs in this work?  Of course, time not considered, the listener would like to continue receiving new information until the end.  That seems like optimal policy.  So whether the listener conveys to the speaker what it would like to see, or if the speaker already has a priority order in which it would reveal / detail more parts of the image, that's not a hugely important distinction in my mind, unless the speaker and listener have very different perceptual abilites, or goals in mind.  It would of course be good to be user-centric in many cases, but how important is it?  I would have liked to see a comparison.

- In comparison to existing work, and bearing in mind the emphasis on human interpretability in the paper narrative, I would have liked to see this method compete with [13]/[14] with a human substituting as the listener, or at least trying to solve the task (and perhaps no communicating).  Without being able to play with the models directly, the previous work seems more interpretable with fewer strokes.  I really find it surprising that humans aren't involved in the measuring of human interpretability, and I think that fact hints that there may be a more suitable name for what is being measured.

Overall I think there is some promising work in how the task is setup, but deviations from the sketch model and the region-based (rather than complexity-based) of existing work seem like steps backwards.  No direct comparisons to previous work, or adaptations of existing work to this, is detrimental both to placing it in the larger research context, and understanding the relative strengths/weaknesses of the proposed approach.

Other comments:

Paragraph 1: These claims seem speculative / opinion-based.

**Questions:**

N/A

---

> ### Author Rebuttal · Authors · 2023-08-10
>
> W1:
>
> First, our work falls into the topic of emergent communication. Emergent communication aims to facilitate human-like communication between intelligent agents, as delineated in [1,2,3,4]. In this work, we propose a novel question answering setting to promote multi-round, bilateral, reciprocal communication between a pair of a sender and a receiver, which clearly follows the definition of emergent communication. See more details in General Response 1.
>
> Second, our motivation is to promote multi-round interaction, and our experiments can support this motivation from two aspects:
>
> 1) Quantatively, Fig 3 shows the downstream performance as a function of the communicaton complexity. We see that the 2-round communication significantly outperforms the 1-round communication across a vast communication complexity. This validates that multi-round interaction promotes higher communication quality.
>
> 2) Qualitatively, Fig. 4 and Fig. 7 illustrates the step-by-step process of the interactive communication. We see that our feedback mechanism enables the sender to pay more attention to the specific areas according to different questions from the receiver, guiding the communication to focus on question-critical area. The experimental findings validate that our multi-round setting simulates human-like multi-round conversation.
>
> ---
>
> W2:
>
> - We agree with you that the evaluation of downstream tasks is important, but only relying on downstream tasks would cause non-interactive and non-interpretable issues; see more details in General Response 1. In comparison, the triangular evaluation is a more comprehensive evaluation metric, see more details about the triangular evaluation in General Response 1.
> - We are not sure how the reviewer defines whether a protocol is contrived or not. But our task setting is more natural and better at promoting emergent communcation than previous works, because it can deliver a higher resemblance to human communication in the real world. Human interactions go far beyond simple games, such as classifying images; more often, they involve bilateral, interative communication to achieve a complicated task. From this perspective, our multi-round question-answering task is more closely aligned with daily human interactions than the common Lewis' game in previous works.
> - Our work provides three interesting insights: the prerequisite, the promoter and the condition of interaction in communication. see more details in General Response 2.
>
> ---
>
> W3:
>
> We simplified the drawer for two reasons.
> - First, existing stroke-based non-iterative drawing method cannot produce sketches with sufficient quality to satisfy the basic needs in ISQA. The sketches provided by [13] can only contain a blurry outline of the major object and can not contain enough information to solve question like "What page is displayed on the monitor?"; see table 1 in [13]. Only iterative methods such as [40], have yielded visually satisfactory results. However, iterative methods are time-consuming, often requiring several minutes to generate a single sketch.
> - Second, our motivation focuses on multi-round interaction, which is an orthogonal direction to drawing. our work emphaisze multi-round interaction, which establises a mechanism that both sender and receiver should be capable to send informative message to each other and guide the direction of the communication. The proposed multi-round interaction framework is also compatible with stroke-based drawing methods. Once a more mature stroke-based drawing method is developed, we can substitute the drawing module in the current framework and test its communication quality.
> Besides, if we consider the AI agents as an intelligence who lives in a discrete world, it is reasonable to consider dot as AI agents' stroke.
>
> ---
>
> W4:
>
> I guess you mean "why generating". Intuivetively, Different questions would require different visual information to provide the corresponding answers, leading to diverse sketches. This is actually one important insight we learnt from experiments. For example, in Figure 7, a representative image is subjected to two distinct questions. The first inquiry, "How many people are visible in the image?", inherently emphasizes the concept of "people". Consequently, regions within the sketch depicting individuals take precedence, as they are more pertinent to the posed question, commanding greater weight in the question-answering module based on the attention network. Conversely, when the question shifts to "Is it still snowing in the image?", the focus broadens to a more global perspective. This is attributed to the absence of a salient region garnering specific attention within the question-answering module for this particular question.
>
> ---
>
> W5:
>
> - Yes, the feedback mechanism between [14] and this work is significantly different; see more details in General Response 3.
> - If we understand the reviewer correctly, the reviewer is considering a comparison between sender-centric and the receiver-centric communication. However, different from these two settings, our setting considers a bilateral, reciprocal communication. The sender and the receiver have equal rights and collaboratively finish the QA task. The sender and the receiver share the same target while neither of them has fully integrated information to reach the target.
>
> ---
>
> W6:
>
> - Our motivation is to promote multi-round, bilateral interaction for emergent communication, instead of creating better drawing merely from the sender perspective; also see W3.
> - We cannot directly compare our work with [13,14], because they are targeting different tasks. Even [13] and [14] have not been directly compared. In this case, a forceful comparison is unfair and cannot provide any useful conclusion.
> - Human experiments: We conduct two human experiments to validate 1) human can be a good receiver (but slightly worse than our model); and 2) CLIP distance and human perception are consistent; see General Response 4.

---

### Author Rebuttal · Authors · 2023-08-10

**General Response 1: Uniqueness of the proposed ISQA task for emergent communication:**

We propose a novel task setting, whose goal is to promote multi-round, bilateral, interactive communication between a pair of a sender and a receiver. To achieve this goal, our proposal has three unique characteristics: information disparity,  triangle evaluation, and feedback mechanism.
- Information disparity. Our setting simulates the question answering between two people. Without communication, neither the sender nor the receiver can accomplish this task alone. Specifically, the sender has the source information (image), but does not know the target information (question); and the receiver has the target information (question), but does not know the source information (image). Therefore, such an information disparity creates a necessity to communicate. Note that most previous works [1, 13, 14] do not create such an information disparity.
- Feedback mechanism. We promote bilateral communication by allowing the receiver to provide feedback to the sender. This simulates multi-round conversation between two people. With the feedback mechanism, both sender and receiver are equal rights to send informative message to each other and guide the direction of the communication. Note that most previous works predominantly concentrate on single-round, sender-receiver architecture. Therefore, we extend emergent communication into a multi-round setting, seeking insights into the impact of the feedback mechanism for the emergent communication process.
- Triangle evaluation. We ensure high-quality communication by designing three criterion. Given the criteria of the ability to handle downstream tasks, the communication content between the sender and the receiver has to be informative. Given the criteria of communication complexity, the communication content has to be compact. Given the criteria of human interpretability, the communication content has to be intuitive. These three factors together promote high-quality communication.

---

**General Response 2: Insights brought by the proposed ISQA task:**

This work provides three insights about how interaction emerges in communication.
  - First, information disparity is the prerequisite for interaction. Fig. 7 shows the feedback messages for the same image according two different questions. We see that our feedback message transfers question-related information to the sender, enhancing communication efficiency via a gradient-informed region of interest (can be displayed as sketch as shown in Figure 2), which enables a human-like multi-round interaction for the first time in visual emergent communication. This provides an insight that one of the reasons why interaction emerges might be querying task-aligned information when receiver is more acquainted with the object than sender.
  - Second, the constrain of communication complexity promotes interaction.  Fig.3 shows the ISQA performance comparison between 2round and 1round communication. We see that the downstream performance can be boosted when interaction is introduced to the visual communication pipeline. This provides a insight that the complexity constrains promote the agents to pursue a more efficient communication leads to interaction.
  - Third, it requires sufficient input information for the feedback to be functional. Fig 3 shows that functional feedback from the receiver relies on adequate communication complexity. This offers the crucial realization that precise feedback hinges on a foundational shared understanding.

---

**General Response 3: Comparison with [14]:**

- Methodologically, the way to design the feedback mechanism is significantly different. The feedback in [14] is a binary signal (yes or no), indicating the receiver's confidence level in rendering a decision. In comparison, the feedback in our work is a region of interests, pointing out which parts of the sketch should get higher attention in the next round.
- Quantatively, the communication quality between [14] and our method can not be directly compared, because the aims and task settings are different and incomparable. [14] focuses on the generation and evolution of pictography and the task it leverages is traditional Lewis' game to create symbols for each class. It is not designed to solve our visual question answering task and cannot execute the visual question answering task. In turns, our work cannot execute Lewis' game as well. So far, there has been no precedents to compare results across different tasks in emergent communication.

---

**General Response 4: Human Experiments to verify CLIP measurement:**

To verify the correlation between CLIP and manual measurement, we conduct a human evaluation, where 12 participants were tasked with assessing the human-interpretability of three different models. Participants were presented with three corresponding images from each model simultaneously and asked to score them based on judgment criteria ranging from 1 to 5 (lower number indicating better interpretability); see Table.1 in rebuttal PDF. We sampled 3,000 sets of images consistent with the settings in Table 1 of the paper. Each set comprised images generated by the Pragmatic, PragGeo, and Geometric models from the same RGB images.

Table.2 in rebuttal PDF compares human interpretability score and CLIP distance. We see that: 1) the results in human experiments and the CLIP distance are consistent to show the order of Geometric, PragGeo Pragmatic in terms of human interpretability; 2) both PragGeo and Geometric models provide an average human interpretability score between partially understand and mainly understand while the sketches provided by Pragmatic models can not be understood; and 3) Comprehensively considering with Fig.3, the PragGeo achieves a more optimal balance among the task performance, drawing complexity and human interpretability.

---

> ### Comment · Area_Chair_JCyp · 2023-08-19
>
> Dear reviewers,
>
> Please read all the other reviewers' discussions and the authors' feedback. Please take a moment to acknowledge the authors' rebuttal and update your rating accordingly.

---

### Decision · Program_Chairs · 2023-09-21

**Decision:**

Accept (poster)

**Comment:**

The paper introduces an innovative approach to multi-round sketch-based emergent communication through the Interactive Sketch Question Answering (ISQA) task. The proposed task explores a novel direction in the field and has garnered interest from the reviewers due to its unique setup and potential contributions.

The concept of multi-round communication, as demonstrated by the ISQA task, presents an exciting avenue for emergent communication research. The reviewers' expertise in the field is evident, and their feedback is based on a solid understanding of the topic. The reviewers' assessments collectively highlight both the strengths and potential limitations of the paper.

Strengths identified by the reviewers include the task's novelty, the integration of sender-understandable feedback (bounding boxes), and the introduction of a triangular evaluation method that takes into account task accuracy, drawing complexity, and human interpretability. The presentation of the paper is generally clear and effective in conveying the problem and proposed solution.

While the paper's contributions are acknowledged, some concerns and suggestions have been raised. Reviewer feedback highlights the need for additional quantitative analysis to establish correlations between automatic and manual measures of communication quality, and the potential to provide more comprehensive comparisons to previous work. Some reviewers also express curiosity about the insights gained from the proposed approach.

Addressing the concerns raised by the reviewers and expanding the analysis could further strengthen the paper's contributions and impact. The novelty of the ISQA task, the balance between interoperability and pragmatism, and the potential to enhance multi-round communication are all promising aspects of this work.

In conclusion, the paper's innovative approach to multi-round emergent communication, coupled with the reviewers' positive perceptions of its potential contributions, suggests that it should be accepted. Revisions guided by the reviewers' feedback will be valuable to ensure that the paper's insights and contributions are well-founded and effectively communicated.